# End-to-end Multiple Instance Learning for Whole-Slide Cytopathology of Urothelial Carcinoma

**Joshua Butke**[1,2]                                    JOSHUA.BUTKE@RUHR-UNI-BOCHUM.DE
[1] *Bioinformatics Group, Department for Biology and Biotechnology, Ruhr-University Bochum, Germany*
[2] *Center for Protein Diagnostics, Ruhr-University Bochum, Germany*

**Tatjana Frick**[2,3]                                    TATJANA.FRICK@RUHR-UNI-BOCHUM.DE
[3] *Chair of Biophysics, Department for Biology and Biotechnology, Ruhr-University Bochum, Germany*

**Florian Roghmann**[4]                                  FLORIAN.ROGHMANN@ELISABETHGRUPPE.DE
[4] *Department of Urology, Ruhr-University Bochum, Marien Hospital Herne, Germany*

**Samir F. El-Mashtoly**[2,3]                         SAMIR.EL-MASHTOLY@RUHR-UNI-BOCHUM.DE
**Klaus Gerwert**[2,3]                                    KLAUS.GERWERT@RUHR-UNI-BOCHUM.DE
**Axel Mosig**[1,2]                                        AXEL.MOSIG@RUHR-UNI-BOCHUM.DE

**Editor:** TBA

## Abstract

As a non-invasive approach, cytopathology of urine sediment is a highly promising approach to diagnosing urothelial carcinoma. However, computational assessment of the cytopathological status of a sample raises the challenge of identifying few cancerous cells among thousands of cells in a microscopic whole-slide image. To address this challenge, we propose an end-to-end trainable multiple instance learning approach that combines the attention mechanism and hard negative mining to classify hematoxylin and eosin stained patient-level whole-slide images of urine sediment cells. The singular cells are extracted by a simple foreground detection algorithm. With feature embeddings computed for each image patch in a bag by a convolutional neural network, the attention mechanism serves as the pooling operator, enabling a bag-level prediction while still giving an interpretable score for each image patch. This enables the identification of *key instances* and potential regions of interest that trigger a patient-level decision. Our results show that the proposed system can differentiate between normal and cancerous urothelial cells, thus enabling the non-invasive diagnosis of urothelial carcinoma in patients using urine sediment analysis.

**Keywords:**   multiple instance learning, urine sediment, cytopathology

## 1. Introduction

Bladder cancer is the 11[th] most commonly diagnosed cancer in the world, with approximately 75% of patients suffering from non-muscle invasive bladder cancer (NMIBC). As bladder cancer is likely to recur, patients often undergo follow-up treatment which is one of the reasons it leads to the highest cost of care per cancer patient (Antoni et al., 2017; Lee et al., 2012).

Cystoscopy and urine cytology still form the backbone of diagnosis and follow-up tests of bladder cancer. Despite the now predominant use of flexible endoscopes, cystoscopy remains

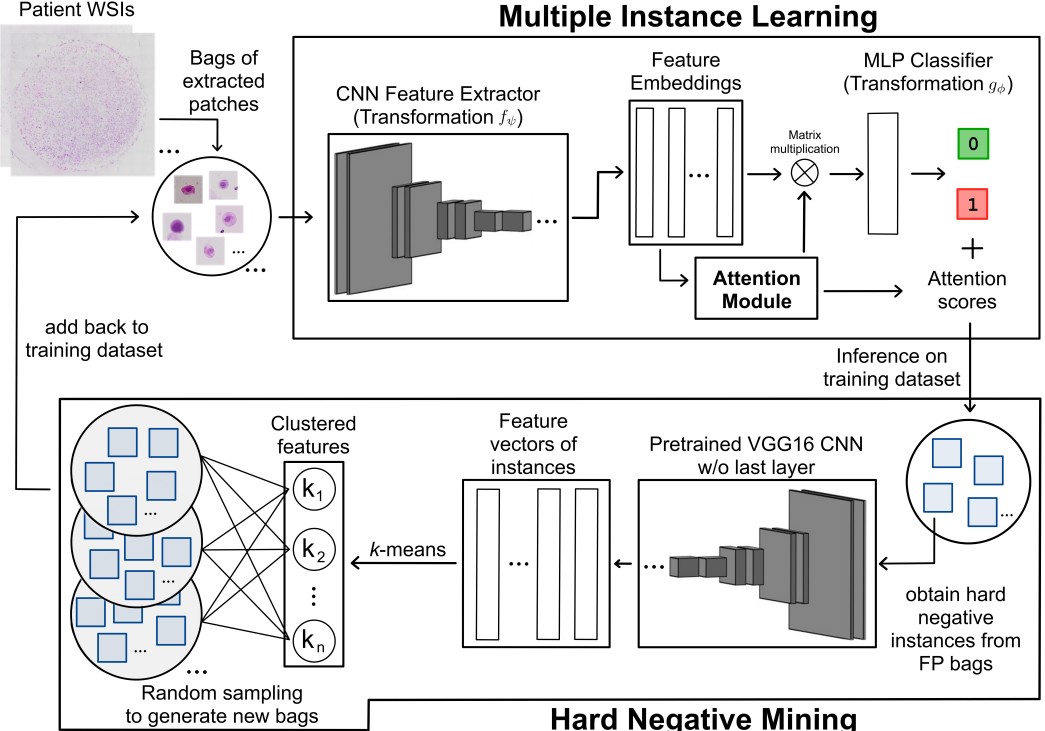

Figure 1: Overview of our proposed approach, integrating attention-based multiple instance learning and hard negative mining for whole-slide cytopathology. Bags of image patches are extracted from WSIs. Each bag passes through a custom CNN feature extractor resulting in a bag classification of positive or negative and the attention scores per instance. Based on these, hard negative instances are collected from false positive bags to generate new bags by hard negative mining.

an invasive diagnostic procedure that is perceived as uncomfortable by patients. The usage of non-invasive alternative diagnostic methods remains challenging as urine cytology for instance is highly dependent on the expertise of the examiner and has limited sensitivity (Sullivan et al., 2010). Several urine-based tests have been developed, but none of them have been recommended in the European Association of Urology (EAU) guidelines for diagnosis and treatment of bladder cancer, and there is consensus that markers can not replace urethrocystoscopy. However, non-invasive methods promise potential support in tumor surveillance through guiding the use of more invasive techniques (Babjuk et al., 2013).

The commonly accepted working hypothesis of urine based urothelial cancer cytopathology is that cancerous urothelial cells are contained in the urine sediment (Kirkali et al., 2005). Yet, the relatively few cancerous urothelial cells are largely outnumbered not just by normal urothelial cells, but also by cells of other types such as squamous cells or leukocytes. Thus, the application of deep learning approaches inevitably faces similar problems

as gigapixel whole-slide images (WSIs) in histopathology, where systematic annotation is feasible only at the whole-sample level, while annotations with cellular or even pixel precision are prohibitively expensive. Yet, WSI based approaches promise improved diagnostics and better treatment strategies compared to the subjective bias introduced among observers and alleviate laborious annotation efforts by trained experts. One potential approach for this setting was prominently introduced by Campanella et al. (2019) in the application of multiple instance learning (MIL) to WSI analysis in the context of histopathology. Here, each WSI is weakly annotated with a coarse-grained patient-level label into a binary classification setting of healthy vs. cancerous. The WSI is then regarded as a bag of extracted or tiled image instances whose individual sub-labels are unknown. A bag is classified as either positive if it contains at least one positive element, or as negative if it contains none.

In our MIL setting, we classify patient urine sediment samples into two disease statuses without any given labeling except for the clinically attributed patient status of urocystitis (negative) or cancerous (positive). Compared to MIL in histopathology, cells in cytopathological WSIs are randomly distributed and thus lack dependencies on their spatial surrounding.

To summarize, the main contributions of our paper are:

- the first approach that works fully end-to-end on whole-slide images of urine sediment;

- the combination of gated attention pooling (Ilse et al., 2018) and a modified version hard negative mining (Li et al., 2019); and

- the use of large, adaptively extracted patches as a crucial step for WSI-based learning.

## 2. Methods

Given a patient WSI of H&E stained urine sediment cells (as a bag $\mathbf{X_i}$) with its extracted instances $X_{ij} \in \mathbf{X_i}, j = 1, ..., N_i$ we try to identify the class label $C_i \in \{0, 1\}$ and potentially the key instances, with $c_{ij} = 1$ of singular urothelial cells in this bag that triggered this decision. Therefore we train an end-to-end deep learning system based on multiple instance learning principles with a trainable pooling operator in the form of the attention mechanism. To improve performance, instances falsely classified as positive are collected to form additional data samples via hard negative mining. The complete, integrative approach can be examined in Fig. 1.

### 2.1 Preprocessing (Extraction of image patches)

There exist many suitable off-the-shelf algorithms for single cell detection to extract suitable image patches from a WSI, such as using the Mask R-CNN (He et al., 2017) architecture in a similar experimental case by Sadafi et al. (2020) or the generalist segmentation approach Cellpose (Stringer et al., 2020). However, using these methods effectively often requires additional training or transfer learning and some amount of professionally annotated data, especially when dealing with the medical imaging domain.

Instead, we propose a simple image binarization based approach, based on Otsu's method (Otsu, 1979) and morphological closing to segment objects in the foreground, find the contours of potential objects and determine their area and centroids by using spatial image moments (Hu, 1962). By empirically setting a lower and upper boundary on the desired area

size, suitable image patches of cells can then be extracted from the collection of centroids per WSI.

## 2.2 Multiple instance learning

In the classic, binary supervised learning setting a model is trained to predict the value of a target categorical variable $c \in \{0, 1\}$ for a given sample or instance $\mathbf{x} \in \mathbb{R}^D$. However, in the case of MIL, instead of a single instance, there exists a so-called bag of instances $X = \{x_1, ..., x_N\}$, where the total number and ordering of instances does not matter and with only a singular label $C$ associated with the bag. According to the original problem formulation by Dietterich et al. (1997) we can assume that individual labels exist for each of the instances in a bag, and while these are not directly accessible we can formulate the MIL problem setting as:

$$C = \begin{cases} 0 & \text{if } \sum_n c_n = 0, \\ 1 & \text{otherwise} \end{cases} \tag{1}$$

To implement MIL there is usually a distinction between two different approaches, instance- and embedding-based. In either way, the model $\Theta(X)$ is composed of, a transformation $f_\psi$ of individual instances, a permutation invariant pooling function $\sigma$ such as mean, max or noisy-or that aggregates all elements in a bag (Wang et al., 2019; Maron and Lozano-Pérez, 1998) and lastly a transformation $g_\phi$ of combined instances (Zaheer et al., 2017) to classify the whole bag such that:

$$\Theta(X) = g_\phi(\sigma(f_\psi(x_1), ..., f_\psi(x_N))) \tag{2}$$

Both transformations can be modeled using neural networks with parameters $\psi$ and $\phi$. In the instance-based approach however transformation $g$ is the identity function. Importantly, both approaches have disadvantages as the embedding-based approach lacks interpretability, while the instance-based approach is hard to train due to error propagation (Liu et al., 2012).

## 2.3 Attention-based pooling

The attention mechanism is widely known and used in a plethora of deep learning tasks. Originally introduced by Bahdanau et al. (2014) it is now a key component in many state-of-the-art applications from natural language processing (Vaswani et al., 2017) to semantic image segmentation (Chen et al., 2016). Typically, it is assumed that all instances are sequentially dependant, however in the case of MIL it was introduced by Ilse et al. (2018) that the attention mechanism can be used as a permutation-invariant pooling operator, substituting the normally used mean or max aggregation operators.

This attention-based pooling calculates a weighted average over the previously extracted feature embeddings, where the weights are learned by a neural network. In our work we employ a slightly modified version, which was also introduced in the same paper of Ilse et al. (2018), called gated attention, as results have shown increased performance in similar histopathological problem settings. Here, an additional, learnable non-linearity in the form of the sigmoid function $\text{sigm}(\cdot)$ is introduced as a gating mechanism, to counter the potential approximately linear behaviour of the $\tanh(\cdot)$ function for $x \in [-1, 1]$ (Dauphin et al., 2017).

Let $H = \{h_1, ..., h_N\}$ be a bag of $N$ instance embeddings as a result of transformation $f_\psi$, then the attention-based MIL pooling is defined as:

$$\sigma_{\mathbf{atten}} = \sum_{i=1}^{N} a_i \mathbf{h}_i, \tag{3}$$

where

$$a_i = \frac{\exp\{\mathbf{w}^\top(\tanh(\mathbf{Vh}_i^\top) \odot \mathrm{sigm}(\mathbf{Uh}_i^\top))\}}{\sum_{j=1}^{N_i} \exp\{\mathbf{w}^\top(\tanh(\mathbf{Vh}_j^\top) \odot \mathrm{sigm}(\mathbf{Uh}_j^\top))\}} \tag{4}$$

in which $\mathbf{w}$, $\mathbf{V}$ and $\mathbf{U}$ are parameters that are learned during training, in contrast to the previously mentioned pooling operators that are not trainable.

The resulting attention scores $a_i$ have numerous advantages, as they provide a scalar, interpretable value for each instance in a bag and act as a similarity measure to compare instances against each other. Additionally the attention mechanism serves as a gradient update filter during training (Ilse et al., 2018), as instances with higher correlated attention will contribute more to learning the model. This methodology extends the embedding-based MIL approach and integrates desirable advantages of the instance-based approach, enabling the discovery of key instances.

### 2.4 Hard negative mining

One further possible use of the learned attention scores is that they can be used to select hard negative instances, which enables us to extend the model with a hard negative mining (HNM) loop (Li et al., 2019). HNM can be used to repeatedly bootstrap negative samples, which are selected from the samples in the training dataset that are falsely classified as positive by the diagnostic model (Dalal and Triggs, 2005). This simple but effective technique is now commonly used in many applications in the medical domain to improve performance on difficult negative instances such as in breast cancer WSI classification (Bejnordi et al., 2017).

We follow the approach proposed by Li et al. (2019) to incorporate a HNM system into our experiments. Given the training bags $\mathbf{X}$ and learned attention scores $\mathbf{a}$ we select all $l$-th bags that have $C_l = 0$ but were predicted as being positive after training. From these false positive bags $\mathbf{F_L}$ we then extract hard negative instances $M$ by attention score filtering:

$$M_l = \{a_{l_i} | a_{l_i} \geq \bar{a}_l + \frac{\sigma_l}{5}\} \tag{5}$$

where $\bar{a}_l$ is the mean and $\sigma_l$ is the standard deviation of the attention scores of the $l$-th bag. The collected hard negative examples can then be used to generate new hard negative bags which are mixed into the training dataset for a second training run.

The generation process is based on a feature extraction stage and a sampling stage. For feature extraction, each hard negative instance in $M$ is passed through a VGG16 deep convolutional neural network, pretrained on the ImageNet dataset (Simonyan and Zisserman, 2014), without its last softmax classification layer, resulting in vectors of size 4096. These feature vectors are then clustered with the $k$-means algorithm (Lloyd, 1982) into $k$ clusters.

To enable the MIL model to better learn features comprehensively, a varied selection of instances into the new bags is used. Thus, in the following sampling stage, the $h$-th new bag $B_h$ is generated by randomly selecting instances from the clusters with a discrete uniform probability distribution. The size of each new bag is limited, based on $s_{min}$ and $s_{max}$, which are the minimum and maximum number of elements found in the original training bags, having a normal distribution based random size, with mean $\mu$ and standard deviation $\sigma$ of all original training bag sizes.

## 3. Experiments

**Dataset**  Cells are stained with hematoxylin and eosin (H&E) staining, before obtaining all whole-slide images with an Olympus BX61VS slide scanner, using the Olympus UPlanSApo 20x/0.75 NA objective lens. For details on urine sampling and preparation of urine sediment cells, we refer to a previous publication (Krauß et al., 2018).

The dataset consists of 40 individual patients, with only a binary disease status (cancer vs normal) given as their annotation. The distribution of classes is imbalanced with 27 patients exhibiting cancer and 13 patients of normal controls with urocystitis, an inflammation of the bladder. Each WSI is approximately 20000x20000 pixels in size, with a resolution of 115 nm/pixel, containing hundreds and thousands of individual cells of different cell types present in urine sediment.

**Implementation Details**  We follow previous works (Ilse et al., 2018; Li et al., 2019; Sadafi et al., 2020) and train the model in a similar fashion, using the Adam optimizer (Kingma and Ba, 2014) with default hyperparameter settings of $\beta_1 = 0.9$, $\beta_2 = 0.999$ and a learning rate of 0.001 for 150 epochs, where the model state at the epoch with the lowest training loss is chosen for validation. For experiments with HNM a second training run with 100 epochs is conducted on the artificially enlarged training dataset. In experiments without HNM the previously saved model state is trained for another 100 epochs on the original dataset to ensure fair comparison.

The model is composed of a deep convolutional neural network feature extraction stages that operates on bags of instances with arbitrary size, the attention-based MIL pooling and a final classification multilayer perceptron. The computed embedding $H$ is multiplied with the just-in-time generated attention matrix $a$ and finally classified on the bag-level. The same training parameters were used across all experimental runs.

Previous publications dealing with MIL and attention often use datasets that lend themselves to very small patches, of up to 32x32 pixels. Our extracted patches of urothelial candidate cells are of size 150x150 pixels with an average bag size of 300 instances. As such the approach is implemented using model parallelism with a batch size of 1, putting the first convolutional layer of the MIL model on its own GPU and then subsequently increasing the amount, using 4 NVIDIA V100 GPUs in total. Further details as well as the code can be found publicly available at https://github.com/butkej/MIL4Cyto.

**Baselines**  To compare the performance of our proposed MIL system with attention and hard negative mining data augmentation loop, we choose different baseline approaches in the form of the two most often encountered pooling operators mean and max, as well attention-based pooling without further dataset augmentation. All baselines are embedding-based

Table 1: Performance evaluation of different methods on our dataset. The average of five runs is reported with its resulting standard error. AUC: area under the receiver operating characteristic curve; FPR: false positve rate; (gat.) atten.: (gated) attention-based pooling; HNM: hard negative mining

| Method | Accuracy | $F_1$ score | AUC | FPR |
|---|---|---|---|---|
| MIL+mean | $0.72 \pm 0.054$ | $0.83 \pm 0.026$ | $0.64 \pm 0.054$ | $0.83 \pm 0.200$ |
| MIL+max | $0.73 \pm 0.057$ | $0.83 \pm 0.026$ | $0.58 \pm 0.090$ | $0.77 \pm 0.230$ |
| MIL+atten. | $0.83 \pm 0.074$ | $0.86 \pm 0.102$ | $0.86 \pm 0.048$ | $0.35 \pm 0.134$ |
| Li et al. (2019) | $0.90 \pm 0.027$ | $0.92 \pm 0.019$ | $0.89 \pm 0.028$ | $0.19 \pm 0.080$ |
| **MIL+atten.+HNM** | $\mathbf{0.92 \pm 0.022}$ | $\mathbf{0.94 \pm 0.016}$ | $\mathbf{0.92 \pm 0.024}$ | $\mathbf{0.20 \pm 0.046}$ |
| **MIL+gat. atten. +HNM** | $\mathbf{0.94 \pm 0.028}$ | $\mathbf{0.96 \pm 0.020}$ | $\mathbf{0.96 \pm 0.025}$ | $\mathbf{0.13 \pm 0.088}$ |

Figure 2: Exemplary single cells found in negative and positive bags and their corresponding attention range. High values of attention occur for archetypal instances such as clearly distinguishable urothelial cells. Low attention values are correctly attributed to other cell types or artifacts that do not inform the bag label.

MIL neural networks. Additionally we test against the best performing configuration of Li et al. (2019), which uses a marginally altered attention mechanism with adaptive weighting to explicitly enlarge differences between positive and negative instances.

**Metrics** We report the accuracy, $F_1$ score, area under the ROC Curve, and false positive rate averaged across 5 runs to compare the different approaches.

### 3.1 Results and Discussion

We perform 3-fold stratified cross validation, in which three independent models are trained and their results are averaged. The data split is performed patient-wise on the full WSIs and resulting bags, in the same way for each run and method to enable fair comparison. In Table 1 we summarize the performance of five different approaches on our dataset. Additionally, Fig. 2 shows the exemplary results of attention scores for different types of cellular objects encountered in positive and negative bags. Similar to its original introduction, the gated attention mechanism performs slightly better than normal attention.

Naive mean and max pooling perform the worst, often only predicting the majority class, thus also having a high FPR. Using the attention-based pooling already performs remarkably better, as urothelial cells which are responsible for the bag label are attributed with high attention. The additional usage of HNM then further reduces the FPR by constructing new hard negative training examples from normal urothelial cells that were attributed a higher than average attention score compared to other instances in their bag. Other cell types and image artifacts that occur in extracted image patches are correctly attributed with low attention as they do not inform the bag label decision. However, using the setting of Li et al. (2019) results in significantly weaker performance. Including softsign activation + adaptive weighting is seemingly detrimental in our problem setting. Instead our approach focuses on a modified selection of pseudo hard negative instances during HNM which achieves better results.

Finding the *key instances* that trigger the bag label is also highly important in the context of interpretability, which is an essential requirement of the currently evolving regulatory standards for *good machine learning practice* (Abels et al., 2019).

## 4. Conclusion

Our results demonstrate reliable classification of patient-level disease status as well as the differentiation between normal and cancerous urothelial cells present in inflammations of the bladder or NMIBC. The explicit identification of relevant cells along with a score of attentiveness provides a degree of interpretability that could be a key for cytopathology as a clinical decision support. This could be strengthened further by incorporating uncertainty estimates, e.g. by utilizing recent Bayesian dropout ideas (Gal and Ghahramani, 2016). In terms of clinical relevance, our approach can be extended to facilitate multi-class differentiation, e.g. for grading or staging of tumors, possibly in combination with a multi-label setting regarding different cell types on the instance level (Feng and Zhou, 2017).

We have introduced a variant of multiple instance learning that improves the performance of related recent approaches by a significant margin. Compared to other MIL cytopathology approaches, our contribution avoids the reliance on precise cell identification, and uses a rather rough background separation and obtains bags of instances by centroid detection. As our results clearly indicate, the combination of attention-based multiple instance learning with hard negative mining fully compensates the coarse grained instance identification, leading to a significant edge over the recently proposed MIL+attention as our baseline method.

### Acknowledgments

This work was supported by the Ministry for Culture and Science (MKW) of North Rhine-Westphalia (Germany) through grant 111.08.03.05-133974 and the Center for Protein Diagnostics (PRODI).

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

## Appendix A. Supplemental Material

### A.1 Urine sampling

Urine samples were collected from patients diagnosed with bladder cancer and also from patients with pathologically confirmed urocystitis but without bladder cancer at the Department of Urology of the Ruhr University Bochum, Marien Hospital Herne, Germany. Institutional review board approval (IRB 3674-10) and written informed consent from all patients were obtained.

### A.2 Methods - Baselines

For the methods of MIL + mean as well as MIL + max the same approach as in our proposed contribution was performed. Due to the missing attention scores no hard negative mining was used. The MIL is based on the embedding approach and uses the same MLP classifier after the pooling function $\sigma$. For the mean pooling we thus have

$$\sigma_{mean} = \frac{1}{N} \sum_{i=1}^{N} \mathbf{h}_i \tag{6}$$

and for the max pooling equivalently

$$\forall_{j=1,\ldots,N_i} : \sigma_{max} = \max_{i=1,\ldots,N}\{\mathbf{h}_{i,j}\} \tag{7}$$

Without the elementwise multiplication term the gated attention mechanism, becomes the standard attention mechanism, which is used in the MIL + attention method:

$$a_i = \frac{\exp\{\mathbf{w}^\top \tanh(\mathbf{Vh}_i^\top)\}}{\displaystyle\sum_{j=1}^{N_i}\exp\{\mathbf{w}^\top \tanh(\mathbf{Vh}_j^\top)\}} \tag{8}$$

in which $\mathbf{w}$ and $\mathbf{V}$ are parameters that are learned during training.

### A.3 Experiments - Implementation details

For all implementations and experiments Pytorch 1.6.0 was used as the main framework with additional usage of scikit-learn 0.23.2 and their respective dependencies. Preprocessing was performed with OpenCV 4.4.0. Further details as well as the code can be found publicly available at https://github.com/butkej/MIL4Cyto.

