# OpenReview forum: "End-to-end Multiple Instance Learning for Whole-Slide Cytopathology of Urothelial Carcinoma"
_MICCAI.org/2021/Workshop/COMPAY — COMPAY 2021_

### Official Review · Reviewer_jVfF · 2021-08-02
**proposes a small, incremental improvement, very small dataset**

**Rating:** 6
**Confidence:** 3

**Review:**

The authors evaluated MIL with attention and hard negative mining to classify cancerous urothelial cells from whole slide images.
Contribution is the combination of attention with hard negative mining (hnm) with some adaptation.
The authors show that hnm and attention together are crucial for the task
Weaknesses: very small dataset with only 40 cases. With 3fold validation the validation set is probably only around 13 cases large and an accuracy improvement of 2% corresponds to being better on one third of a single case on average. Future work will require much more evaluation. Still, the results seem consitent.
It is also unclear whether the cnn feature extractor and the feature extractor from hnm are different, and if so, why.
Recommendation: weak accept

---

### Official Review · Reviewer_5k9U · 2021-08-19
**Review by 5k9U**

**Rating:** 8
**Confidence:** 3

**Review:**

The paper proposes an end-to-end trainable multiple instance learning approach that combines the attention mechanism and hard negative mining to classify H&E stained patient-level whole-slide images of urine sediment cells. Overall, this is a good paper tackling an important clinical problem of assisting the interpretation of cytopathology of urine sediments, which can be an important step forward for bladder cancer prognosis. It would be worth including the runtime characteristics of the MIL approach proposed, particularly with respect to the sensitivity to the patch generation steps from the WSI images. This would boost the clinical relevance of the proposed ideas. Additionally, the translational scope of the proposed approaches merits some discussion as well. The sensitivity of the hard negative mining steps with respect to the margins employed also merits an ablation study, as does the influence of the backbones used for the MIL steps (the CNN feature extractor) and the HNM steps (extending to backbones beyond pretrained VGG 16?). Overall, the manuscript has solid clinical relevance and can only be improved further with larger datasets (possibly multi-center and multi-scanner), sensitivity testing over other backbones, optimization of computational complexity and inference time, and a reporting of statistical significance of the results obtained.

---

### Decision · Program_Chairs · 2021-08-25

Accept